# Improving Shelf Life and Content of Unsaturated Fatty Acids in Meat of Lambs Fed a Diet Supplemented with Grape Dregs

**DOI:** 10.3390/foods12234204

**Published:** 2023-11-22

**Authors:** Yali Yao, Hongbo Wang, Zhenzhen Lu, Fang Nian, Chen Zheng, Fadi Li, Defu Tang

**Affiliations:** 1College of Animal Science and Technology, Gansu Agricultural University, Lanzhou 730070, China; 18419166373@163.com (Y.Y.); Luzz@st.gsau.edu.cn (Z.L.); nianf@gsau.edu.cn (F.N.); zhengc@gsau.edu.cn (C.Z.); 2Laboratory of Quality & Safety Risk Assessment for Livestock Products, Ministry of Agriculture, Lanzhou Institute of Husbandry and Pharmaceutical Sciences, Chinese Academy of Agricultural Sciences, Lanzhou 730050, China; wanghongbo@caas.cn; 3College of Pastoral Science and Technology, Lanzhou University, Lanzhou 730070, China; lifd@lzu.edu.cn

**Keywords:** grape pomace, meat production, shelf life, CLA deposit-related genes

## Abstract

This study was conducted to evaluate the potential effects of dietary grape residue levels on the slaughter indicators, meat quality, meat shelf-life, unsaturated fatty acid content, and expression of fatty acid deposition genes in the muscle of lambs. Sixty 30-month-old male Dorper and Small-Tailed Han F1 hybrid lambs were assigned to a single factor complete randomized trial design and fed with four different diets including 0%, 8%, 16%, and 24% grape dregs, respectively. The findings regarding meat production efficacy in the lambs revealed substantial differences. The control group showed notably lower dressing percentage, carcass weight, net meat weight, meat percentage concerning carcass, meat-to-bone ratio, relative visceral and kidney fat mass, and rib eye area compared to the other groups (*p* < 0.05). Additionally, the meat shearing force of lambs fed a diet with 16% grape pomace (GP) was significantly higher than that of the 24% GP group (*p* < 0.05), while the 24 h meat color parameter a* value of the control group was notably higher than that of the 8% GP group (*p* < 0.05). In addition, compared to the control group, lambs fed with a diet containing 16% GP had higher levels of oleic acid (C18:1n-9c), linoleic acid (C18:2n-6c), behenic acid (C22:0), tricosanoic acid (C23:0), lignoceric acid (C24:0), and conjugated linoleic acid (CLA), at a ratio of ∑CLA/TFA, ∑n-6, ∑MUFA, and ∑PUFA in the *longissimus dorsi* muscle (*p* < 0.05), but the reverse case was applicable for Total Volatile Basic Nitrogen (TVB-N) content (*p* < 0.05). GP supplementation did not substantially affect the expression of *stearoyl-CoA desaturase* (*SCD)*, *peroxisome proliferator activated receptor alpha* (*PPARα*), and *peroxisome proliferator-activated receptor gamma* (*PPARγ*) genes (*p* > 0.05). The findings indicated that incorporating grape dregs in the diets of fattening lambs leads to notable enhancements in meat production and the antioxidant capacity of lamb meat, and effectively extends the shelf life of the meat.

## 1. Introduction

The nutritional richness of mutton, including its high content of protein, iron, zinc, and vitamin B, is gaining recognition and preference among people [1]. In recent years, as the overall living standards of the general population have notably improved, consumers have increasingly prioritized the freshness of various mutton products. “Shelf life” has emerged as a crucial criterion for assessing mutton freshness. However, in actual production, the limited shelf life of fresh (frozen) meat has become a significant issue. A shorter shelf life can result in increased surface water loss from meat products, accelerating meat deterioration and consequently reducing the storage duration of these products [2,3]. Lipid oxidation-induced protein oxidation of meat products has been identified as the main chemical factor affecting the storage time of meat products [4]. Lipid oxidation could significantly shorten the shelf life of meat products, resulting in poor color, flavor deterioration, water retention, and tenderness [5].

Several studies have shown that the supplementation of grape seed [6], grape residue concentrate [7,8], and high tannin sorghum [9] to animal diets can effectively prevent the lipid oxidation in mutton and substantially prolong the shelf life of mutton by enhancing the activity of antioxidant. For instance, the addition of 20% grape dregs to the lambs’ diet can improve both the meat productivity and meat shelf life of lamb without affecting the sensory quality [10]. Furthermore, incorporating grape dregs into pig diets could enhance meat quality, growth performance, and nutrient utilization, and also lead to changes in the fatty acid composition of subcutaneous fat [11]. Interestingly, Valenti et al. [12] reported that a diet containing 4% condensed tannin can reduce the content of monotrans 18:1 fatty acids in lamb muscle. Hence, incorporating a specific quantity of grape dregs in the diet can enhance the quality of sheep meat by influencing the composition of fatty acids.

The factors that influence the fatty acids in the meat are primarily the hardness of adipose tissue, shelf life (oxidation of lipids and pigments), and the composition of aromatic substances [13,14]. The fatty acid and lipid composition of beef has been found to play a crucial role in metabolism and can ultimately affect meat quality [15]. Stearoyl CoA desaturase (*SCD*) serves as a crucial enzyme capable of catalyzing the creation of double bonds from both saturated and unsaturated fatty acids. It plays a pivotal role in regulating the internal production of conjugated linoleic acid (CLA) in animals [16]. When trans oleic acid is absorbed into the animal body, it can be converted into c9 and t11-CLA through a dehydrogenation reaction by △-9 dehydrogenase (*SCD*) in the liver and mammary microsomes [17]. The results indicated that 78% of c9 and t11-CLA in tallow were mainly derived from the Δ-9 dehydrogenation of t11-oleic acid [18]. Diacylglycerol acyltransferase (DGAT) stands as a pivotal enzyme in triglyceride synthesis. In the mammary glands of cows, DGAT activity displayed a higher preference for oleyl (c9 18:1) coenzyme A compared to c9, t11 C18:2, t10, c12 C18:2 coenzyme A, and c9, c12 C18:2 coenzyme A. This preference could potentially lead to a reduction in triglyceride content in the milk [19]. Interestingly, this preference does not impact the transcription of DGAT1 and DGAT2 genes. However, it is suggested that prior to transcription, it might potentially regulate the translation of CLA isomers or influence their enzymatic activities. [20]. Moreover, it was found that peroxisome proliferator-activated receptor (*PPAR*) can affect the expression of *SCD* gene [21].

Building on prior research findings, this study sought to enhance the quality of lamb meat, prolong its shelf life, and modify the composition and proportion of fatty acids or CLA by introducing grape residue into the diets of fattening lambs. Additionally, we examined the potential impact of grape dregs on the expression of genes associated with CLA deposits, including *SCD*, *PPARα*, and *PPARγ* in lamb tissues. These findings hold significant promise for the future utilization of grape pomace in both the food and feed industries.

## 2. Materials and Methods

All samples were collected strictly following the ethical code (GSAU-Eth-AST-2022-035) approved by the Animal Welfare Committee of Gansu Agricultural University.

### 2.1. Materials

Fresh grape dregs (the ratio of grape skin to grape seed is 1:1.22) were obtained from Shiyanghe Winery of Minqin in Gansu Province, and were dried naturally, crushed, and used in the experiments after passing through a 0.5 mm sieve.

### 2.2. Experimental Design

In this work, 60 two-month old weaned, healthy, and identical weight Dorper sheep
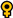
 × Small-Tailed Han sheep
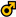
, F1 hybrid (24.28 ± 2.50 kg, *n* = 15) were selected in a single factor complete randomized trial design. Following the principle of equal weight distribution, 15 sheep in each group were evenly allocated into 4 dietary treatment groups. These groups were fed diets comprising 0%, 8%, 16%, and 24% grape dregs, respectively. After measuring the dry matter, crude protein, calcium, phosphorus, and condensed tannin in the grape residue of feed raw materials, the trial diet formula was developed according to 90% of the nutrient requirements for fattening of NRC (2007) commercial lambs (Table 1).

### 2.3. Sample Collection and Measurement Parameters

#### 2.3.1. Slaughter

After 60 days of feeding, 10 lambs whose body weight was close to the average weight were selected from each group. After 24 h of fasting and 12 h of drinking, the lambs were electrocuted (200 V voltage was applied for 4 s) and slaughtered. The measurements recorded after the slaughter included carcass weight, bone weight, net meat weight, carcass net meat percentage, dressing percentage, rib eye area, growth rate (GR) value, as well as weights of visceral fat, kidney fat, tail fat, and dorsal fat in accordance with the methodology recommended by Orzuna-Orzuna et al. [22].

#### 2.3.2. Meat Quality

The left *longissimus dorsi* muscle of lambs was collected and placed into a 4 °C refrigerator to determine both meat quality and shelf life following slaughter. The pH value was measured with a pH meter (testo205, Shanghai testo instruments international trade Co., Ltd., Shanghai, China) after 45 min and 24 h. The water loss rate was measured with a WW-2A strain gauge unconfined force meter. The cooked meat rate and dripping loss were analyzed based on a method recommended by Orzuna-Orzuna et al. [22]. The meat color was determined by Minolta CR-400 color meter (Chroma meter, CR-410, Tokyo, Japan) after the meat was stored in 4 °C refrigerator for 1–2 h or for 24 h. Tenderness was assessed using a C-LM tenderness meter, with three replicates conducted for each sample.

#### 2.3.3. Determination of the Relevant Indicators of Shelf Life

Approximately 300 g samples of the *longissimus dorsi* muscle of each sheep were collected within 1–2 h following slaughter. Following division into 11 distinct portions, they were vacuum-packed and stored in a refrigerator at 4 °C for subsequent testing. Meat color, pH, and total volatile basic nitrogen (TVB-N) were measured at intervals of 0, 1, 2, 3, 4, 5, 6, 7, 8, 9, and 10 days. The analysis of volatile basic nitrogen was conducted following the method outlined in the Analysis of Health Standards for Meat and Meat Products (GB/T5009.44-2003). Each sample was tested in triplicate.

#### 2.3.4. Determination of the Muscle Fatty Acid Content

The observations indicated that the diets containing 16% and 24% grape residue effectively extended the shelf life of the meat. However, lamb meat fed with 24% grape residue exhibited a notably higher drip loss compared to those fed with 16% grape residue. Therefore, the control and 16% GP groups were selected to study the fatty acid content and expression of various genes related to fatty *longissimus dorsi* acid deposition. Approximately 50 g of the right *longissimus dorsi* was collected from the lambs of the control and 16% GP groups, which were vacuum packed and stored at −80 °C for determination of fatty acid as well as CLA.

Fatty acid extraction: First, an appropriate amount of sample was weighed and ground into a powder using liquid ammonia. The powder was then placed into a hydrolysis tube, followed by the addition of 5.3 mL methanol (Bioteke, Beijing, China) and 0.7 mL 10 mol/L KOH solution (Bioteke, Beijing, China). The mixture was shaken vigorously until the sample was completely immersed in the liquid. It was then put into a 55 °C water bath for 1.5 h, shaken briefly every 10 min, and allowed to cool. Next, 0.6 mL of 24 mol/L H_2_SO_4_ (Bioteke, Beijing, China) was added and kept in a 55 °C water bath for another 1.5 h, shaking briefly every 15 min, followed by cooling to room temperature. Subsequently, 3 mL of n-hexane was added, and the solution was vortexed for 5 min before it was transferred to a 15 mL centrifuge tube. After centrifugation at 1500 r/min for 5 min, 1 mL of the supernatant was collected and analyzed using gas chromatography–mass spectrometry (Scion 456-GC, Bruker, Billerica, MA, USA).

Chromatographic conditions: The analysis utilized an SP2560 chromatographic column (100 m × 0.25 mm × 0.20 µm). The injection port temperature was set at 220 °C, utilizing a 9:1 split flow mode for injection. The temperature program for the chromatographic column proceeded as follows: the initial column temperature was maintained at 120 °C for 5 min. It was then increased to 200 °C at a rate of 3 °C/minute and held for 10 min. Finally, the temperature was elevated to 240 °C at a rate of 1.5 °C/min, resulting in an operational time of 78.333 min. Helium was employed as the carrier gas.

Mass spectrum conditions: In the analysis, the full scan mode was utilized with a 9 min solvent delay. The gain factor set was 10. The ion source temperature was maintained at 230 °C, with a maximum of 250 °C. The quadrupole temperature was set at 150 °C, with a maximum of 200 °C.

#### 2.3.5. Detection of Gene Expression

##### Total RNA Extraction and cDNA Synthesis

Following slaughter, samples of *longissimus dorsi* muscle, back fat, and liver from 10 lambs in groups A and C were immediately collected, placed in liquid nitrogen, and then sent back to the laboratory for storage at −80 °C for the total RNA extraction. TaKaRa MiniBEST Universal RNA Extraction Kit (Code: No.9767) was utilized to extract RNA from the sheep tissues (operated according to the instructions provided in the kit). The RNA concentration and purity were determined with an ultramicro spectrophotometer (Nano Drop 2000, Thermo Scientific, Waltham, MA, USA). This test analysis indicated varying extraction efficiencies of genomic RNA in different tissues. However, the concentration consistently exceeded 80 ng/mL. The extracted RNA exhibited an OD260/OD280 ratio ranging between 1.8 and 2.0, and an OD260/OD230 ratio of around 2.0, meeting the prerequisites for subsequent analyses. To assess RNA integrity, 1% agarose gel electrophoresis with EB staining was performed. Under ultraviolet light, bright 28s and 18s RNA bands were observed, with the former being twice as bright as the latter, indicating the absence of sample degradation (see Figure 1). The reverse transcription process was conducted using the PrimeScriptTM RT reagent Kit (ThermoFisher Scientific, Rockford, IL, USA) following the provided instructions within the kit. This facilitated the synthesis of single-stranded cDNA, with three replicates performed per group.

##### Design and Synthesis of the Primers

Sheep △9 dehydrogenase (stearoyl-CoA desaturase, *SCD*) gene, peroxisome proliferator activated receptor alpha (*PPARα*), peroxisome proliferator-activated receptor gamma (*PPARγ*), and glyceraldehyde-3-phosphate dehydrogenase (*GAPDH*) mRNA sequences were detected by NCBI. The primers for these genes were designed by PrimerPremier5 primer synthesis software using their gene mRNA as templates and synthesized by Dalian Bao Biological Engineering Co., Ltd., Dalian, China. The primer sequences of the target gene and internal reference gene have been shown in Table 2.

##### Real-Time Fluorescence Quantitative PCR Reaction

Before the Light Cycler^®^ 480 Real-Time PCR System (Roche, Indianapolis, IN, USA) reaction, all the genes were tested with conventional PCR primers to verify their amplification products. For the determination of *SCD*, *PPARα*, and *PPARγ* gene expression, a 10 μL real-time PCR reaction system was prepared according to the specifications outlined in Table 3. The reaction procedure involved a 30 min pre-denaturation at 95.0 °C. The PCR reaction was conducted with the following steps: an initial 5 s reaction at 95.0 °C, followed by a 30 min reaction at 54.5 °C, board reading. A total of 40 cycles were completed. The solution curve was analyzed within the temperature range of 65.0 °C to 95.0 °C, with plate readings taken at intervals of 0.5 °C (reading the plate after maintaining a constant temperature for 5 s). Three replicates were performed for each sample, and the average value was then calculated.

### 2.4. Statistical Analysis

According to the Ct value obtained by the real-time fluorescence quantitative PCR method, 2^−ΔΔCT^ was used to measure expression of *SCD*, *PPARα,* and *PPARγ* genes. The test data were analyzed with an independent samples t-test and One-Way ANOVA using SPSS23.0 statistical software, and were expressed as $\bar{X}\pm SE$. *p* < 0.05 indicated significant differences. When the difference was significant, Duncan’s method was used for multiple comparison.

## 3. Results

### 3.1. Effect of Grape Residue Content in the Diet on Lamb Meat Production

The potential effects of GP levels in diets on lamb meat production are shown in Table 4. The observations revealed notable differences among the groups: lambs fed a diet with 8% grape pomace (GP) exhibited significantly higher carcass net meat percentage, rib eye area, growth rate, and dorsal fat relative mass compared to the control group (*p* < 0.05). Additionally, the carcass weight and net meat weight of the 16% GP group surpassed those of the other groups (*p* < 0.05). Interestingly, lambs on a diet containing 24% GP showed a higher slaughtering rate, bone-to-meat ratio, and relative mass of visceral, kidney, and dorsal fats compared to the other groups (*p* < 0.05).

### 3.2. Effect of Dietary Grape Residue Content on Lamb Meat Quality

The effects of dietary GP levels on lamb meat quality are shown in Table 5. The shear force of lambs fed with a diet containing 16% GP was significantly higher than that of 24% GP groups (*p* < 0.05). The 24 h meat color parameter a* value of the control was higher than that of the 8% and 16% GP groups (*p* < 0.05).

### 3.3. Effect of Grape Residue on Shelf Life of Lamb Meat

#### 3.3.1. Effect of Grape Residue on pH of Lamb Meat during Storage

The effects of dietary GP levels on lamb meat pH are shown in Table 6. The pH of lambs fed a diet containing GP was not altered significantly.

#### 3.3.2. Effect of Grape Residue on Meat Color Parameters of Lamb Meat during Storage

The effects of dietary GP levels on the various flesh parameters of lamb meat are shown in Table 7. The values of a* (day 7, 9, 10), b* (day 4), and L* (day 7) of lambs fed a diet containing GP were significantly higher than those of the control (*p* < 0.05).

#### 3.3.3. Effect of Grape Residue Content in the Diet on TVB-N Content in Lamb Meat during Storage

The effects of dietary GP levels on TVB-N of lamb meat are depicted in Table 8. It was found that on the day of lamb slaughter, the TVB-N values of lambs fed with a diet containing 8% and 16% GP groups were significantly lower than that of the control (*p* < 0.05). However, the TVB-N of 16% and 24% GP groups was higher than those of the 8% GP group on the first day of the storage (*p* < 0.05). The content of TVB-N of the control and 8% GP group lamb stored in vacuum packaging in a refrigerator at 4 °C for 0–7 days was less than 15 mg/100 g, as specified in the national GB2707-2016, and after the 8th day, the content of TVB-N exceeded the value specified in the national standard. The content of TVB-N in 16% and 24% GP group lamb stored in a refrigerator at 4 °C for 0–8 days was found to be below 15 mg/100 g, as reported in the national GB2707-2005, and after the 9th day, the content of TVB-N exceeded the value specified in the national standard.

### 3.4. Effect of Grape Residue Content in Diet on Fatty Acid Composition and Content of Longissimus Dorsi Muscle in Lambs

The effects of dietary GP levels on lamb meat fatty acid content in the *longissimus dorsi* muscle are shown in Table 9. The C18:1n-9c, C18:2n-6c, C22:0, C23:0, C24:0, and CLA in the ratio of ∑CLA/TFA, ∑n-6, ∑MUFA, and ∑PUFA of lambs fed with a diet containing 16% GP were significantly higher than that of the control (*p* < 0.05).

### 3.5. Impact of Fodder Grape Slag Content on the Lamb SCD, PPARα, PPARγ Gene Expression in Different Organizations

As depicted in Figure 2, mRNA expression levels of *SCD* in the liver, *longissimus dorsi* muscle, and dorsi fat of lambs in the experimental group supplemented with 16% GP were 94.87%, 61.64%, and 121.29% higher than that in the control group, respectively. The mRNA expression of *PPARα* in the liver and *longissimus dorsi* muscle of the experimental group fed 16% grape residue was observed to be 18.75% and 13.49% higher, respectively, compared to the control group. Meanwhile, the expression of the *PPARγ* gene in the same experimental group was notably higher, showing a 196.89% increase in the liver and a 28.43% increase in the *longissimus dorsi* muscle compared to the control group.

## 4. Discussion

### 4.1. Effect of Grape Residue Content on the Meat Productivity and Fat Distribution of Lambs

The important indexes for measuring animal growth and slaughtering are dressing percentage and net meat weight. In this study, the dressing percentage of lambs fed diets containing 8%, 16%, and 24% GP was 48.84%, 48.77%, and 49.18%, respectively, which was significantly higher than that of the control and was consistent with the report of Tian et al. [23]. However, Yagoubi et al. [24] have previously reported that the dressing percentage and net meat percentage of Dorper and Small-Tailed Han F1 hybrid lambs were 49% and 80%, respectively, which were higher than those observed in this work. It was also found that the carcass weight decreased substantially when the sheep were fed diets containing tannins [25], and it was inferred that the thin carcass of sheep could be due to an improved protein content of carcass. Nevertheless Priolo et al. [26] hypothesized that feeding tannins to lambs could lead to malnutrition of lambs. The study noted a significant increase in the carcass weight of lambs from the 16% and 24% grape pomace (GP) groups compared to the control group. This enhancement might be attributed to the potential protective effects of the condensed tannins present in grape dregs on the protein in the feed used in lamb diets. This phenomenon potentially aids in effectively increasing the proportion of feed protein that passes through the rumen, subsequently promoting the absorption and utilization of feed protein by the lambs [10].

The results indicated that the rib eye area, GR value, relative mass of the visceral fat, kidney fat, and dorsal fat in lambs fed with a GP-containing diet were significantly higher than those of the control, and the rib eye area of lambs in this experimental group was also remarkably higher than that of the lambs reported in the previous study [6], suggesting that the use of a diet supplemented with GP promoted the performance in terms of both meat production and fat deposition in lambs.

### 4.2. Effect of Dietary Grape Residue Content on Lamb Meat Quality and Shelf Life

Interestingly, several previous studies have reported that some characteristics related to the meat quality, such as pH_45min_, L*, b*, drip loss, cooking loss, and marbling score, were not affected by dietary supplementation of grape seed extract (GSPE) [27], but dietary GSPE inclusion may affect the muscle glycolysis potential and lactate content during the postmortem period because the muscle glycolysis potential is negatively correlated with the final pH value [28], whereas the drip loss is positively correlated with the muscle lactate content [28,29]. While the results from this test show some inconsistency compared to previous tests, it is notable that the pH value remained within the acceptable range of 5.6 to 5.8 [30], excluding the possibility of the formation of dark, solid, and dry meat (secretly cut) or stress in lambs [31]. Therefore, it is speculated that the addition of dietary glucose could effectively regulate the expression of glucose transporter 4 (GLUT4) in the insulin-mediated muscle glycogen conversion process and further influence the muscle glycolysis process [32]. Our findings indicate that incorporating 16% grape dregs into the feed notably raised the muscle’s shear force. This increase may be linked to the elevated final pH value resulting from the grape dregs’ treatment, considering the reported inverse relationship between the final pH value and meat tenderness [33].

In addition, the change in meat color is closely linked to lipid oxidation and hemoglobin acts as the catalyst for lipid oxidation [34]. Grape dregs contain highly condensed tannins and the antioxidant activity of tannins is four times that of vitamin E [35]. The microbial community within the rumen can significantly influence the antioxidation process in the animal’s body [36]. For instance, Toral et al. [35] confirmed that rumen microorganisms were unable to degrade the condensed tannins and produce inertia in the digestive tract, but caused chemical modification in the intestines, and were further depolymerized as well as decomposed into small molecules to be absorbed through the intestines. Condensed tannins can change the fatty acid composition in the meat, thus influencing the oxidation process and achieving an antioxidant effect [37]. Mancini et al. [38] reported that the addition of tannin to rabbit feed decreased the brightness L* value. Research also indicated that supplementing lamb feed with tannins led to heightened meat color values for a* and b*. Moreover, there was a positive correlation found between meat color b* values and the degradation of meat color, while meat color a* values showed a negative correlation with this degradation process [34]. The effect of tannic acid on the meat color is unclear, but it may be related to the increase in iron content in the muscles as tannin levels increase. The results demonstrated that the meat color L* value of the groups containing 16% and 24% grape residue on the day of slaughter was significantly higher than that of the control group, and after 24 h, the meat color L* and a* values of the * value decreased, but the b* value increased. The brightness of the 8% GP group showed an initial increase from day 1 to day 5, followed by a decline in the brightness L* value starting from day 6 onwards. On the other hand, the 16% GP group displayed flesh color brightness L* values between 40 and 45 from day 1 to day 7, with a subsequent decline from day 8 onward. These findings suggest that GP can significantly reduce myoglobin content in the meat, consequently altering its color and brightness [34].

Previous studies have shown that lipid peroxidation and reduced meat color in ruminant meat are primarily affected by fatty acid composition and antioxidants present in the tissues [39]. It was noted that beef fed with forage and cereal feed maintained the bright red color associated with oxygenated myoglobin for the longest duration throughout the retail period [40], and that oregano with antioxidant properties improved the color stability of bison steaks without any negative impact on their sensory attributes [41]. The findings of this work indicated that incorporating 16% grape pomace enhanced the stability of meat color in lamb, aligning with previous test outcomes. This improvement might be attributed to grape pomace slowing down the pH drop process. Rapid pH decline triggers acid reactions with muscle proteins, causing protein denaturation and oxidation, ultimately altering the meat’s color, freshness, and overall quality [42,43]. Meanwhile, TVB-N has been considered as the standard of freshness of meat and fish in China as well as most other countries, and it is an important index for evaluating the shelf life of meat [44]. The national standard GB2707-2005 has updated the acceptable Total Volatile Basic Nitrogen (TVB-N) content from 20 mg/100 g to 15 mg/100 g. Fresh meat registering TVB-N levels below 15 mg/100 g is considered safe and suitable for consumption. During its shelf life, if the TVB-N content exceeds 15 mg/100 g, the meat is categorized as spoiled and not fit for consumption [44]. However, in actual production, the short shelf life of fresh meat is now a major problem. It was found that from the 0th day to the 7th day of storage, the TVB-N content of lambs fed diets containing 0% and 8% GP was remarkably lower than 15 mg/100 g, which was in line with the national standard, whereas by the 8th day and thereafter, the TVB-N content was measured to be 15.92 mg/100 g and 21.27 mg/100 g, respectively, exceeding the 15 mg/100 g threshold, indicating that the meat was no longer fresh. However, the TVB-N content remained below 25 mg/100 g, classifying it within the secondary meat category without significant deterioration. Additionally, from day 0 to day 8, the TVB-N content of lamb diets containing 16% and 24% GP consistently remained below 15 mg/100 g. On the 9th day and after the storage, the TVB-N content of 16% and 24% GP groups was 15.94 mg/100 g and 21.65 mg/100 g, respectively. The shelf life of the 16% and 24% GP groups was found to be one day longer than that of the control. Thus, lamb diets containing 16% and 24% GP can effectively prolong the shelf life of meat.

### 4.3. Effect of Dietary Grape Residue Content on Fatty Acid Composition and Content of Lamb Longissimus Dorsi

There has been only limited research on the impact of condensed tannins on fatty acids in lamb, particularly regarding the influence of grape dregs on lamb fatty acids. Tannins play a significant role in altering fatty acid deposition within the rumen’s biohydrogenation pathway [45]. In the rumen biohydrogenation process, C18:2n-6 and C18:3n-3 are gradually hydrogenated to C18:0, whereas the intermediates in the biohydrogenation process are CLA and t-11C18:1, which can be controlled by cellulose-decomposing bacteria, such as a *Butyrivibrio fibrisolvens* control [46]. Costa et al. [47] believe that tannins can effectively inhibit the activity of rumen microorganisms and reduce the process of rumen biohydrogenation. In this study, we found that the content of ∑CLA in group C with 16% grape residue (15.70 mg/100 g fresh meat) was 62.36% higher in comparison to that in group A (9.67 mg/100 g fresh meat), and the content of CLA-t10 and c12 in group C containing 16% GP (4.09 mg/100 g fresh meat) was 61.66% higher than that in group A (2.53 mg/100 g fresh meat). Hence, incorporating 16% grape dregs into the diet can notably elevate the Conjugated Linoleic Acid (CLA) content in the *longissimus dorsi* muscle of lambs, effectively enhancing the CLA levels in lamb meat. Francisco et al. [39] reported that addition of condensed tannins to the lamb diet increased the content of t-11-18:1 in muscle fat. Vasta et al. [48] found that adding tannins to the rumen fluid significantly decreased the biohydrogenation of linoleic acid and the saturation of t-C18:1, consequently hindering the advancement of biohydrogenation and elevating the concentration of t-C18:1. These results suggested that 16% grape dregs increased the contents of ∑MUFA and ∑n-6, but decreased the contents of ∑n-3. In another study, Priolo et al. [26] completely fed the lambs with grass rich in tannins and showed that the content of C18:3n-3 (linolenic acid) in intramuscular fat of lambs was remarkably higher than that of the control, and the proportion of C20:5n-3 was also increased. In summary, when the grape residue content reaches 16%, it seems to hinder the formation of n-3PUFA but potentially promotes the synthesis of n-6PUFA instead. The n-6/n-3 ratio in the 16% GP group exceeded that of the control (which was 3.03). However, it is worth noting that the n-6/n-3PUFA ratio in this study was lower than the recommended value of 4 suggested by Enser et al. [49].

Vasta et al. [48] found that the content of C18:2n-6 and PUFA in the muscle of lambs fed with a tannin diet (6% DM) was significantly higher than that in the control. In this study, the ∑PUFA content in the 16% GP group measured at 342.71 mg/100 g fresh meat, showcasing a 56.65% increase compared to group A without GP, which registered 218.78 mg/100 g fresh meat. Interestingly, it was observed that when tannins were introduced into the lamb diet, the C14:1 (oleic acid) content significantly surpassed that of the control group [50]. C14:1 is synthesized through the process of muscle endogenesis by the action of *SCD* [51]. Vasta et al. [48] found that tannins can increase the expression of *SCD* in the muscles. In our work it was found the content of C14:1 (carnitine oleic acid) in (4.99 mg/100 g fresh meat) in the 16% GP group was also substantially higher than that of the control (5.45 mg/100 g fresh meat). Therefore, it can be speculated that the condensed tannins contained in the 16% GP group effectively increased the expression of *SCD* in the muscle.

### 4.4. Effect of Dietary Grape Residue Content on the Expression of SCD and PPAR Genes in Lamb Tissues

The mechanism of action of tannins on the *SCD* gene is still unclear. Besharati et al. [52] assumed that the condensed tannins consumed by ruminants are neither degraded nor absorbed in their digestive tract. Previous research has shown that tannic acid found in quebracho is mainly composed of proanthocyanidins. When present in low quantities, these proanthocyanidins can be digested and absorbed [51]. Besharati et al. [52] reported that the bacteria in the rumen and colon can degrade the condensed tannins into several small phenols. Tannins are known to interfere with protein and fatty acid metabolism in the rumen [48]. An indirect effect of tannins on *SCD* gene expression in muscle cannot be excluded. *SCD* can effectively catalyze the endogenous synthesis of MUFA and c9-t-11 CLA [53]. It was observed that an increase in *SCD* expression and its activity can eventually lead to an increase in both MUFA and c9-t-11CLA in ruminant meat. The results revealed that 16% grape residue can significantly increase the expression of *SCD* mRNA in the liver, *longissimus dorsi,* and adipose tissue of lamb, which was consistent with the previous report of Vasta et al. [48]. The mechanisms by which grape residue affects SCD gene expression should, however, be investigated in detail.

*PPARs,* as ligand-activated transcription factors, play a crucial role in governing the expression of various genes implicated in diverse lipid metabolism pathways. These pathways encompass fatty acid transport, intracellular fatty acid binding, degradation (including β-oxidation and ω-oxidation), cellular absorption, and storage of lipids [54]. Currently, the study of the *PPAR* gene in ruminants is predominantly based on *PPARγ*, but there are only a few studies related to *PPARα* reported in the literature. Yang et al. [20] reported that the mRNA expression of the *PPARγ* gene in *longissimus dorsi* muscle was negatively correlated with intramuscular fat content in the sheep. Muhlhausler et al. [55] found that the mRNA expression of the *PPARγ* gene was associated with the fat thickness on the back of lambs. The results showed that the mRNA expression of *PPARα* in both the liver and dorsi fat of the experimental group including 16% GP was 18.75% and 13.49%, respectively, which was significantly higher than that of the control group, and *PPARγ* gene expression was 196.89% and 28.43% higher than that of the control group, respectively. Thus, supplementing feed with grape dregs was observed to enhance the expression of PPAR genes in both the liver and dorsi fat. This finding holds significance for the potential uses of grape pomace extract in the food and feed industry. However, a more detailed exploration is needed to understand the precise regulatory mechanisms behind this effect.

## 5. Conclusions

The inclusion of 16% and 24% grape dregs in the diet notably extended the shelf life of lamb meat. Specifically, the presence of 16% grape dregs significantly boosted Conjugated Linoleic Acid (CLA) levels and the concentrations of various unsaturated fatty acids (C18:1n-9c, C18:2n-6c, C20:3n-6, C20:4n-6, C22:0, C22:1n-9, C22:2), as well as saturated fatty acids (C23:0 and C24:0) in the *longissimus dorsi* muscle of the lambs. This increase ultimately led to a significant rise in the ∑CLA/Total Fatty Acid (TFA) ratio.

## Figures and Tables

**Figure 1 foods-12-04204-f001:**
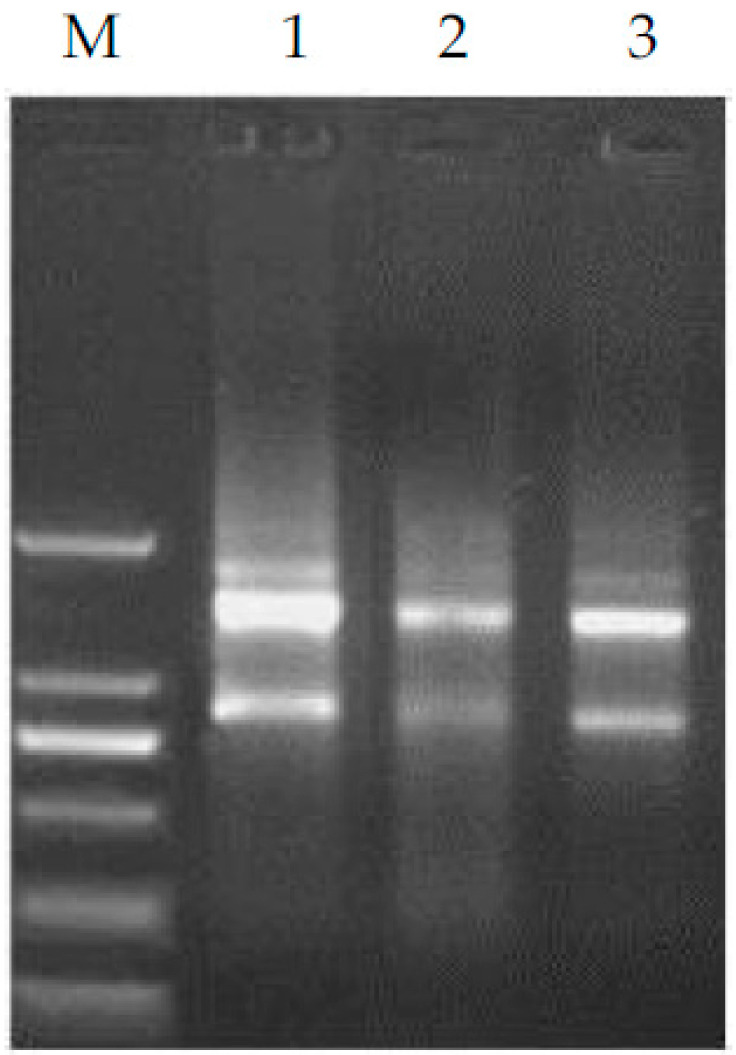
Result of agarose gel electrophoresis of the total RNA obtained from liver, *longissimus dorsi* muscle, and dorsal fat of hybrid lambs. M: marker; 1, 2, and 3 were liver, dorsal fat, and *longissimus dorsi* muscle, respectively.

**Figure 2 foods-12-04204-f002:**
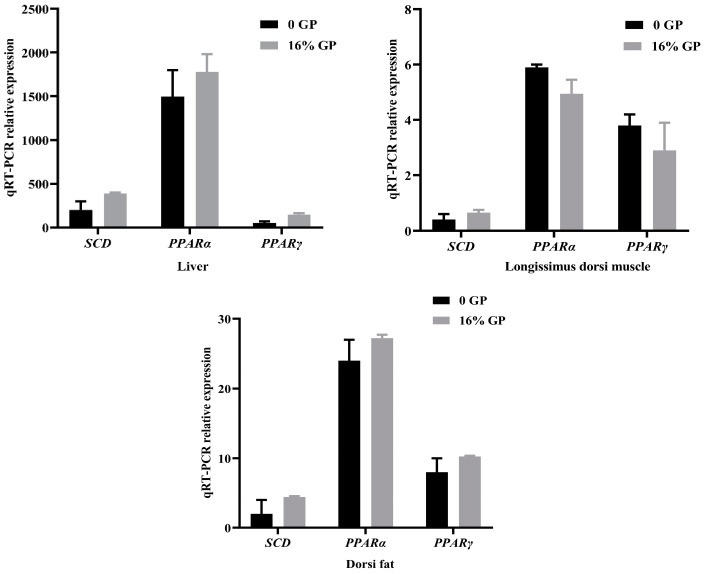
Effect of grape pomace on *SCD*, *PPARα,* and *PPARγ* mRNA expression in liver, *longissimus dorsi* muscle, and dorsi fat tissues.

**Table 1 foods-12-04204-t001:** Composition and nutrient levels of experimental diets (as fed basis, %).

Ingredients	0	8%	16%	24%
Maize	25.5	30.5	31.7	34.5
Barley malt sprouts	3.0	2.0	1.0	1.0
Linseed meal	4.0	2.0	1.0	1.0
Soybean meal	4.0	2.0	2.0	1.0
Cottonseed meal	2.0	2.0	2.0	1.0
Barley straw	44.8	25.3	24.1	14.3
Silage corn	9.5	21.0	15.0	16.0
Grape skins	0	3.6	7.2	10.8
Grape seeds	0	4.4	8.8	13.2
Alfalfa meal	5.0	5.0	5.0	5.0
CaCO_3_	0.2	0.2	0.2	0.2
Ammonium sulfate	0.7	0.7	0.7	0.7
Urea	0.8	0.8	0.8	0.8
NaCl	0.5	0.5	0.5	0.5
Mineral premix *	0.4	0.4	0.4	0.4
Vitamin premix **	0.1	0.1	0.1	0.1
Total	100.0	100.0	100.0	100.0
Nutrition levels				
DE (MJ/kg)	11.20	11.20	11.20	11.20
CP/%	12.00	12.00	12.00	12.00
Ca/%	0.42	0.42	0.42	0.42
P/%	0.20	0.20	0.20	0.20
NDF/%	43.2	42.7	42.0	41.8
Concentrate to forage ratio	4:6	4:6	4:6	4:6
Condensed tannins/%	0	1.5	3.0	4.5

Note: * Mineral premix (mg/kg): S, 200 mg/kg; Fe, 25 mg/kg; Zn, 40 mg/kg; Cu, 8 mg/kg; I, 0.3 mg/kg; Mn, 40 mg/kg; Se, 0.2 mg/kg; Co, 0.1 mg/kg. ** Vitamin premix (IU/kg): VA, 940 IU/kg; VE, 20 IU/kg; VD, 132 IU/kg. DE: dietary energy. CP: crude protein. NDF: neutral detergent fiber.

**Table 2 foods-12-04204-t002:** Primer of *SCD*, *PPARα*, *PPARγ* and *GAPDH*.

Target Genes	Primer Sequences (5′-3′)	Length/bp	GenBank Accession No.
*SCD*	Forward: GGCACATCAACTTTACCACATTCTT	120	NM_001009254.1
	Reverse: TTTCCTCTCCAGTTCTTTTCATCC		
*PPARα*	Forward: GCTTCCACAAGTGCCTTTCC	244	XM_027968213.2
	Reverse: GGCGGATTGTTGTTGGTCTT		
*PPARγ*	Forward: ACGGGAAAGACGACAGACAAA	150	XM_015102096.3
	Reverse: AAACTGACACCCCTGGAAGATG		
*GAPDH*	Forward: AAGTTCCACGGCACAGTCAA	125	NM_001190390.1
	Reverse: ACCACATACTCAGCACCAGC		

**Table 3 foods-12-04204-t003:** Real-time PCR reaction system.

Reagent	Amount
SYBR^®^*Premix Ex Taq*II (Tli RNaseH Plus)	5.0 μL
PCR Forward Primer	0.2 μL
PCR Reverse Primer	0.2 μL
RT reaction mixture (cDNA solution)	0.6 μL
ddH_2_O	4.0 μL
Total	10.0 μL

**Table 4 foods-12-04204-t004:** Effects of GP levels in diets on lamb meat production.

Items	GP Levels				*p*-Value
	0	8%	16%	24%	
Live body weight/kg	27.16 ± 0.40	29.46 ± 0.77	29.85 ± 0.72	28.86 ± 0.82	0.067
Carcass weight/kg	12.60 ± 0.32 ^b^	14.55 ± 0.51 ^a^	14.98 ± 0.45 ^a^	14.76 ± 0.48 ^a^	0.005
Dressing percentage/%	45.71 ± 0.71 ^b^	48.84 ± 0.64 ^a^	48.77 ± 0.43 ^a^	49.18 ± 0.43 ^a^	0.001
Net meat weight/kg	8.69 ± 0.22 ^b^	10.36 ± 0.23 ^a^	10.50 ± 0.54 ^a^	9.56 ± 0.14 ^ab^	0.000
Meat percentage to carcass/%	67.96 ± 0.59 ^b^	70.08 ± 0.69 ^a^	67.66 ± 0.28 ^b^	69.37 ± 0.36 ^ab^	0.009
Meat–bone ratio	2.48 ± 0.08 ^b^	2.67 ± 0.04 ^ab^	2.61 ± 0.09 ^ab^	2.87 ± 0.06 ^a^	0.007
Rib eye area/cm^2^	15.11 ± 0.53 ^b^	18.52 ± 0.97 ^a^	17.28 ± 0.79 ^ab^	17.42 ± 0.71 ^ab^	0.030
Growth rate/mm	7.48 ± 0.38 ^b^	11.34 ± 0.81 ^a^	10.44 ± 0.66 ^a^	10.34 ± 0.73 ^a^	0.004
Relative mass of the visceral fat g/kg	6.10 ± 1.35 ^b^	8.94 ± 1.39 ^ab^	8.40 ± 0.96 ^ab^	11.07 ± 0.86 ^a^	0.041
Relative mass of the kidney fat/%	3.24 ± 0.38 ^b^	3.97 ± 0.47 ^b^	4.22 ± 0.54 ^ab^	4.38 ± 0.61 ^a^	0.004
Relative mass of the tail fat/%	3.38 ± 0.63	4.27 ± 0.72	4.17 ± 0.43	4.38 ± 0.61	0.650
Relative mass of the dorsal fat/%	10.00 ± 0.54 ^b^	14.41 ± 1.17 ^a^	10.99 ± 0.68 ^b^	11.35 ± 0.66 ^b^	0.003

Note: In the same row, different small letters indicate a significant difference (*p* < 0.05), and the same or no letters indicate no significant difference (*p* > 0.05). The same is applied below.

**Table 5 foods-12-04204-t005:** Effects of GP levels in diets on lamb meat quality.

Items	GP Levels				*p*-Value
	0	8%	16%	24%	
pH_1h_	6.94 ± 0.02	6.75 ± 0.02	6.78 ± 0.02	6.92 ± 0.02	0.062
pH_24h_	5.73 ± 0.05	5.58 ± 0.05	5.53 ± 0.09	5.56 ± 0.08	0.153
Water loss rate	6.54 ± 0.29	6.52 ± 0.18	6.50 ± 0.32	6.72 ± 0.34	0.957
Cooked meat rate	71.70 ± 0.57	70.97 ± 0.56	72.00 ± 0.58	73.13 ± 0.72	0.104
Drip loss	2.47 ± 0.14	3.09 ± 0.18	2.70 ± 0.20	2.84 ± 0.14	0.080
Shear force	4.90 ± 0.29 ^ab^	5.38 ± 0.24 ^ab^	5.66 ± 0.15 ^a^	4.71 ± 0.13 ^b^	0.021
L*_1h_	35.04 ± 0.96	34.79 ± 0.25	36.37 ± 0.33	35.16 ± 0.42	0.348
a*_1h_	14.50 ± 0.49	14.05 ± 0.20	14.74 ± 0.34	13.64 ± 0.22	0.095
b*_1h_	6.44 ± 0.21	7.01 ± 0.25	7.13 ± 0.20	6.81 ± 0.11	0.130
L*_24h_	41.15 ± 0.63	41.91 ± 0.45	40.08 ± 0.33	39.92 ± 0.72	0.078
a*_24h_	13.31 ± 0.24 ^a^	11.85 ± 0.22 ^b^	12.90 ± 0.25 ^ab^	13.10 ± 0.33 ^a^	0.013
b*_24h_	11.17 ± 0.49	11.82 ± 0.15	11.55 ± 0.12	11.97 ± 0.23	0.190

Note: In the same row, different small letters indicate a significant difference (*p* < 0.05), and the same or no letters indicate no significant difference (*p* > 0.05).

**Table 6 foods-12-04204-t006:** Effects of GP levels in diets on lamb meat pH.

Items	GP Levels				*p*-Value
	0	8%	16%	24%	
pH_0d_	6.94 ± 0.02	6.75 ± 0.02	6.78 ± 0.02	6.92 ± 0.02	0.062
pH_1d_	5.52 ± 0.06	5.50 ± 0.03	5.54 ± 0.02	5.48 ± 0.04	0.710
pH_2d_	5.65 ± 0.64	5.55 ± 0.09	5.60 ± 0.05	5.51 ± 0.07	0.532
pH_3d_	5.64 ± 0.09	5.70 ± 0.13	5.66 ± 0.13	5.77 ± 0.06	0.841
pH_4d_	6.04 ± 0.09	5.96 ± 0.09	5.80 ± 0.08	6.01 ± 0.08	0.185
pH_5d_	5.82 ± 0.09	5.75 ± 0.07	5.76 ± 0.06	5.57 ± 0.15	0.350
pH_6d_	5.90 ± 0.18	5.88 ± 0.09	5.87 ± 0.07	5.86 ± 0.05	0.993
pH_7d_	5.73 ± 0.10	5.76 ± 0.04	5.70 ± 0.08	5.79 ± 0.08	0.886
pH_8d_	5.84 ± 0.12	5.76 ± 0.28	5.68 ± 0.13	5.42 ± 0.15	0.421
pH_9d_	5.57 ± 0.12	5.65 ± 0.16	5.56 ± 0.09	5.41 ± 0.09	0.530
pH_10d_	5.49 ± 0.15	5.28 ± 0.08	5.27 ± 0.09	5.42 ± 0.12	0.488

**Table 7 foods-12-04204-t007:** Effects of dietary GP levels on lamb meat flesh parameters.

Times	Items	GP Levels				*p*-Value
		0	8%	16%	24%	
0d	L*	35.04 ± 0.96	34.79 ± 0.25	36.37 ± 0.33	35.16 ± 0.42	0.348
	a*	14.50 ± 0.49	14.05 ± 0.20	14.74 ± 0.34	13.64 ± 0.22	0.095
	b*	6.44 ± 0.21	7.01 ± 0.25	7.13 ± 0.20	6.81 ± 0.11	0.130
1d	L*	40.73 ± 1.17	41.32 ± 0.95	42.29 ± 1.19	41.00 ± 0.81	0.757
	a*	14.74 ± 0.25	14.55 ± 0.39	13.92 ± 0.40	14.90 ± 0.50	0.348
	b*	10.92 ± 0.46	12.11 ± 0.63	12.06 ± 0.42	11.28 ± 0.35	0.249
2d	L*	41.90 ± 1.15	43.17 ± 1.20	43.14 ± 0.78	41.81 ± 0.74	0.672
	a*	15.19 ± 0.40	14.88 ± 0.50	14.19 ± 0.57	14.61 ± 0.89	0.388
	b*	13.22 ± 0.65	13.21 ± 0.23	13.15 ± 0.44	12.10 ± 0.62	0.360
3d	L*	41.54 ± 1.02	43.43 ± 0.76	42.39 ± 0.80	43.41 ± 0.64	0.302
	a*	15.57 ± 0.52	15.25 ± 0.25	15.47 ± 0.25	15.81 ± 0.47	0.782
	b*	13.97 ± 0.65	14.37 ± 0.42	14.04 ± 0.74	13.83 ± 0.47	0.933
4d	L*	42.16 ± 0.94	45.22 ± 1.15	43.42 ± 0.74	41.08 ± 0.81	0.053
	a*	15.84 ± 0.23	15.09 ± 0.38	15.34 ± 0.55	14.98 ± 0.22	0.501
	b*	12.38 ± 0.87 ^a^	14.39 ± 0.30 ^a^	14.49 ± 0.36 ^a^	9.58 ± 0.64 ^b^	0.000
5d	L*	42.30 ± 0.86	44.50 ± 0.62	43.13 ± 0.89	43.79 ± 1.15	0.406
	a*	15.17 ± 0.40	14.67 ± 0.34	14.25 ± 0.51	13.82 ± 0.16	0.083
	b*	14.78 ± 0.69	14.29 ± 0.31	14.47 ± 0.61	15.34 ± 0.81	0.671
6d	L*	42.63 ± 0.74	41.53 ± 0.71	42.35 ± 1.16	41.61 ± 1.08	0.816
	a*	14.71 ± 0.70	15.51 ± 0.80	15.89 ± 0.66	13.58 ± 0.42	0.091
	b*	15.25 ± 2.16	15.00 ± 0.71	15.48 ± 2.18	16.87 ± 2.50	0.930
7d	L*	38.01 ± 1.24 ^b^	39.80 ± 0.35 ^ab^	41.52 ± 0.98 ^a^	38.02 ± 0.66 ^b^	0.022
	a*	68.70 ± 0.86 ^a^	68.93 ± 0.42 ^a^	60.14 ± 2.68 ^b^	67.86 ± 1.68 ^a^	0.006
	b*	44.05 ± 4.11	29.69 ± 5.15	30.45 ± 4.67	32.49 ± 5.06	0.188
8d	L*	37.81 ± 1.50	36.80 ± 0.67	37.22 ± 1.44	37.83 ± 0.95	0.900
	a*	68.62 ± 1.24	66.10 ± 2.12	66.70 ± 1.37	66.97 ± 1.06	0.676
	b*	39.99 ± 3.11	37.93 ± 2.15	37.92 ± 2.19	35.59 ± 1.79	0.626
9d	L*	36.09 ± 1.17	38.36 ± 0.98	36.27 ± 0.53	36.98 ± 0.75	0.239
	a*	64.14 ± 1.76 ^b^	68.34 ± 0.66 ^ab^	67.27 ± 0.86 ^ab^	69.25 ± 0.67 ^a^	0.016
	b*	35.04 ± 0.73	38.19 ± 1.48	37.11 ± 1.47	35.81 ± 0.71	0.276
10d	L*	37.93 ± 1.04	35.35 ± 0.76	37.24 ± 0.80	37.17 ± 0.88	0.347
	a*	70.57 ± 1.06 ^a^	65.97 ± 0.62 ^b^	67.13 ± 0.97 ^ab^	69.51 ± 0.89^a^	0.010
	b*	36.32 ± 1.31	36.74 ± 2.02	37.15 ± 1.83	34.89 ± 1.77	0.821

Note: In the same row, different small letters indicate a significant difference (*p* < 0.05), and the same or no letters indicate no significant difference (*p* > 0.05).

**Table 8 foods-12-04204-t008:** Effects of dietary GP levels on lamb meat TVB-N.

Items	GP Levels				*p*-Value
	0	8%	16%	24%	
TVB-N_0d_	10.27 ± 0.48 ^a^	6.73 ± 0.91 ^b^	9.62 ± 1.06 ^ab^	8.92 ± 0.88 ^ab^	0.043
TVB-N_1d_	11.64 ± 1.28 ^ab^	8.97 ± 0.52 ^b^	12.78 ± 0.91 ^a^	13.41 ± 0.98 ^a^	0.020
TVB-N_2d_	10.53 ± 0.74	11.45 ± 0.53	11.28 ± 1.30	9.37 ± 1.45	0.635
TVB-N_3d_	12.99 ± 1.17 ^a^	12.16 ± 0.60 ^ab^	11.49 ± 0.43 ^ab^	10.29 ± 0.26 ^b^	0.026
TVB-N_4d_	12.37 ± 0.52 ^ab^	10.68 ± 0.50 ^b^	14.22 ± 0.96 ^a^	14.72 ± 0.86 ^a^	0.005
TVB-N_5d_	12.92 ± 0.88	11.18 ± 0.95	13.42 ± 0.34	14.11 ± 1.75	0.388
TVB-N_6d_	12.16 ± 1.73	11.94 ± 0.76	14.89 ± 0.10	14.02 ± 0.53	0.484
TVB-N_7d_	13.16 ± 0.66	14.85 ± 0.52	14.22 ± 0.64	12.98 ± 1.45	0.402
TVB-N_8d_	15.92 ± 0.73	16.18 ± 0.80	14.03 ± 1.35	14.50 ± 0.60	0.152
TVB-N_9d_	19.65 ± 1.66	17.91 ± 0.72	19.27 ± 0.74	20.05 ± 0.92	0.484
TVB-N_10d_	21.21 ± 1.85	21.27 ± 2.20	15.94 ± 1.49	21.65 ± 1.62	0.121

Note: In the same row, different small letters indicate a significant difference (*p* < 0.05), and the same or no letters indicate no significant difference (*p* > 0.05).

**Table 9 foods-12-04204-t009:** Effects of dietary GP levels on lamb meat fatty acid content (mg/100 g fresh muscle) in *longissimus dorsi* muscle.

Items	GP Levels		*p*-Value
	0 GP	16% GP	
Total fatty acids (TFA)	2879.26 ± 325.07	4566.13 ± 657.13	0.053
C6:0	0.56 ± 0.02	0.74 ± 0.15	0.294
C8:0	9.92 ± 0.23	10.37 ± 0.32	0.281
C10:0	4.93 ± 0.45	8.85 ± 2.34	0.141
C12:0	10.33 ± 1.39	18.19 ± 4.30	0.124
C13:0	1.05 ± 0.14	1.44 ± 0.32	0.301
C14:0	122.77 ± 20.83	167.90 ± 33.32	0.282
C14:1	5.45 ± 0.70	4.99 ± 0.97	0.713
C15:0	17.90 ± 2.62	27.93 ± 5.46	0.140
C16:0	689.26 ± 80.58	1034.57 ± 151.49	0.080
C16:1	53.35 ± 5.51	66.62 ± 9.33	0.255
C17:0	38.21 ± 5.04	56.16 ± 9.44	0.133
C18:0	622.26 ± 84.46	1088.62 ± 190.93	0.061
C18:1n-9c	986.11 ± 121.46	1691.46 ± 238.43	0.031
C18:2n-6c	137.35 ± 11.09	219.50 ± 21.25	0.008
C18:3n-6	10.52 ± 1.30	12.75 ± 1.45	0.282
C18:3n-3	14.29 ± 1.38	15.74 ± 2.03	0.565
CLA-c9, t11	7.70 ± 1.25	12.95 ± 2.00	0.049
CLA-t10, c12	2.53 ± 0.25	4.09 ± 0.35	0.006
C20:0	5.95 ± 0.81	9.16 ± 1.49	0.096
C20:1	3.52 ± 0.36	4.78 ± 0.71	0.147
C21:0	12.36 ± 0.32	12.79 ± 0.29	0.348
C20:2	0.76 ± 0.08	1.21 ± 0.21	0.084
C20:3n-6	3.67 ± 0.14	5.00 ± 0.14	0.000
C20:4n-6	40.58 ± 1.92	50.90 ± 1.28	0.001
C20:3n-3	0.37 ± 0.03	0.53 ± 0.10	0.156
C20:5n-3	3.08 ± 0.17	3.43 ± 0.20	0.205
C22:0	2.47 ± 0.13	3.93 ± 0.40	0.010
C22:1n-9	1.19 ± 0.09	1.44 ± 0.12	0.107
C22:2	0.87 ± 0.03	1.09 ± 0.07	0.016
C23:0	1.81 ± 0.05	2.13 ± 0.11	0.027
C24:0	11.21 ± 0.39	12.35 ± 0.32	0.048
C22:6n-3	1.93 ± 0.10	2.03 ± 0.11	0.509
C24:1	2.60 ± 0.21	3.11 ± 0.45	0.329
∑CLA	9.67 ± 1.32	15.70 ± 2.67	0.074
∑CLA/TFA	0.32 ± 0.01	0.41 ± 0.03	0.019
∑n-6	54.63 ± 1.86	66.39 ± 1.29	0.000
∑n-3	21.17 ± 2.31	20.96 ± 1.76	0.944
∑SFA	1656.72 ± 169.87	3084.16 ± 624.34	0.064
∑MUFA	1044.28 ± 127.97	1773.39 ± 249.43	0.033
∑PUFA	218.78 ± 15.08	342.71 ± 23.45	0.002
n-6/n-3	3.03 ± 0.21	3.00 ± 0.23	0.910
C16:1/C16:0	0.074 ± 0.001	0.071 ± 0.003	0.227
C18:1/C18:0	1.726 ± 0.058	1.773 ± 0.059	0.570
∑MUFA/∑SFA	0.72 ± 0.01	0.77 ± 0.02	0.053
∑PUFA/∑SFA	0.14 ± 0.01	0.14 ± 0.02	0.879

Note: ∑n-6 fatty acids were calculated as the sum of C18:2n-6c, C18:3n-6, C20:3n-6, and C20:4n-6. ∑n-3 fatty acids were calculated as the sum of C18:3n-3, C20:3n-3,C20:5n-3, C22:5n-3, and C22:6n-3. ∑SFA was calculated as the sum of C6:0, C8:0, C10:0, C12:0, C13:0, C14:0, C15:0, C16:0, C17:0, C18:0, C20:0, C21:0, C22:0, C23:0, and C24:0. ∑MUFA was calculated as the sum of C16:1, C14:1, C16:1, C18:1n-9c, C20:1, C22:1n-9, and C24:1. ∑PUFA was calculated as the sum of C18:2n-6c, C18:3n-3, C18:3n-6, CLA-c9t11, CLA-t10c12, C20:2, C20:3n-3, C20:4n-6, C20:5n-3, C22:2, and C22:6n-3.

## Data Availability

The data used to support the findings of this study can be made available by the corresponding author upon request.

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
