# Peer review of "Improving Shelf Life and Content of Unsaturated Fatty Acids in Meat of Lambs Fed a Diet Supplemented with Grape Dregs"

_foods, 2023, doi:10.3390/foods12234204_

Round 1

Reviewer 1 Report

Comments and Suggestions for Authors

Dear Authors,

Your paper delves into an intriguing subject; however, some revisions are necessary for enhanced clarity and academic robustness.

Title:

Consider refining the title for a more academically appropriate tone. For instance, "Improving Shelf-life and Content of Unsaturated Fatty Acids in Lambs Meat Diet Supplemented with Grape Dregs ."

Introduction:

The introductory section, while well-written, might benefit from a more concise approach. Consider shifting some detailed information to the discussion section for a more focused introduction.

Material and Methods:

Experimental Design: Ensure consistency in notation between Table 1 and the manuscript. Abbreviated terms in the table should be explained in a footnote for better comprehension. Reference the NRC 2007 as a citation source.

Formatting: Utilize italics for "longissimus dorsi" throughout the manuscript for consistency.

Analysis:

Express regret regarding the decision not to conduct all analyses on the four groups, particularly the 16% and 24% grape residue supplemented groups. These omitted analyses could provide valuable insights.

 for every analyses conducted mention the number of observations. n=?

Results and Discussion:

Chapter 3: Restructure Chapter 3 as "Results" rather than "Results and Analysis."

Shelf-life Parameters: Consider revising the focus of pH and color as representations of shelf-life parameters. While pH may indicate freshness, elaborate on how color, if supported by antioxidant compounds, contributes to quality.

Fatty Acids Unit: Ensure the inclusion of the fatty acids unit (mg/100g fresh muscle) in the relevant table.

Discussion Section: The discussion lacks an analysis of the meat's shelf life, an important aspect deserving attention.

Grape Residue Analysis:

Question why the fatty acids profile in grape residue was determined but not the antioxidant capacity or total polyphenol content. Suggest linking these unexplored areas to a comprehensive analysis of the raw feed material.

Your research is interesting, but refining these points will strengthen the paper's academic integrity and broaden its scholarly impact. Please consider these suggestions for the paper's improvement.

Comments on the Quality of English Language

Minor typos and misspelled words. 

Author Response

Dear Editor,

We appreciate you and the reviewers for your precious time in reviewing our paper and providing valuable comments. It was your valuable and insightful comments that led to possible improvements in the current version. The authors have carefully considered the comments and tried our best to address every one of them. We hope the manuscript after careful revisions meet your high standards. The authors welcome further constructive comments if any. Below we provide the point-by-point responses. All modifications in the manuscript have been highlighted in yellow.

Response to Reviewer 1

[General Comment] Your paper delves into an intriguing subject; however, some revisions are necessary for enhanced clarity and academic robustness.

Response: Thank you very much. We will carefully revise and improve it.

Title

Consider refining the title for a more academically appropriate tone. For instance, "Improving Shelf-life and Content of Unsaturated Fatty Acids in Lambs Meat Diet Supplemented with Grape Dregs."

Response: Thanks for your comments. The reviewer offers a more concise alternative to the previous title, so we will directly change the article's title to "Improving Shelf-life and Content of Unsaturated Fatty Acids in Lambs Meat Diet Supplemented with Grape Dregs".

Introduction

The introductory section, while well-written, might benefit from a more concise approach. Consider shifting some detailed information to the discussion section for a more focused introduction.

Response: Thank you very much for the suggestion. The introduction has been simplified for the sake of focus.

Material and Methods: Experimental Design: Ensure consistency in notation between Table 1 and the manuscript. Abbreviated terms in the table should be explained in a footnote for better comprehension. Reference the NRC 2007 as a citation source.

Response: We agree with the suggestions. We have made revisions accordingly. The abbreviations CP, NDF, and DE in the table have been explained in the footnote. Note: DE: dietary energy. CP: crude protein. NDF: neutral detergent fiber.

Formatting: Utilize italics for "longissimus dorsi" throughout the manuscript for consistency.

Response: Thank you very much for the reminder. We have made revisions accordingly throughout the entire text.

Analysis

Express regret regarding the decision not to conduct all analyses on the four groups, particularly the 16% and 24% grape residue supplemented groups. These omitted analyses could provide valuable insights.

Response: Thanks for your kind reminders. In a previous study on meat quality and shelf-life, we observed that diets containing 16% and 24% grape residue were effective in prolonging the shelf-life of meat, but drip losses were significantly greater in lamb fed diets containing 24% grape residue compared to diets containing 16% grape residue. This is supported by who recently recommended a maximum inclusion level of 12% of diet DM after observing quadratic responses in DMI, ADG, and HCW when feeding increasing amounts of grape pomace in lamb diets[1].Therefore, at a later stage, we chose the dietary group containing 16% grape residue and the control group for the fatty acid study.

for every analyses conducted mention the number of observations. n=?

Response: Revised accordingly. It can be seen in line 98, 131, 139 and 190.

Results and Discussion

Chapter 3: Restructure Chapter 3 as "Results" rather than "Results and Analysis."

Response: Revised accordingly. It can be seen in line 224.

Shelf-life Parameters: Consider revising the focus of pH and color as representations of shelf-life parameters. While pH may indicate freshness, elaborate on how color, if supported by antioxidant compounds, contributes to quality.

Response: Thanks for your kind reminders. We have added in 4.2 of the text. It was noted that beef fed with forage and cereal feed maintained the bright red color associated with oxygenated myoglobin for the longest duration throughout the retail period [40], and that oregano with antioxidant properties improved the color stability of bison steaks without any negative impact on their sensory attributes [41]. The findings of this work indicated that incorporating 16% grape pomace enhanced the stability of meat color in lamb, aligning with previous test outcomes. This improvement might be attributed to grape pomace slowing down the pH drop process. Rapid pH decline triggers acid reactions with muscle proteins, causing protein denaturation and oxidation, ultimately altering the meat's color, freshness, and overall quality [42, 43].

Fatty Acids Unit: Ensure the inclusion of the fatty acids unit (mg/100g fresh muscle) in the relevant table.

Response: Thanks for your suggestion, the relevant content have been revised accordingly. It can be seen in line 273 and 275.

Discussion Section: The discussion lacks an analysis of the meat's shelf life, an important aspect deserving attention.

Response: Thanks for your kind reminders. We've already discussed this in line 356 and 377. We have added in 4.2 of the text as “that oregano with antioxidant properties improved the color stability of bison steaks without any negative impact on their sensory attributes [41]. The findings of this work indicated that incorporating 16% grape pomace enhanced the stability of meat color in lamb, aligning with previous test outcomes. This improvement might be attributed to grape pomace slowing down the pH drop process. Rapid pH decline triggers acid reactions with muscle proteins, causing protein denaturation and oxidation, ultimately altering the meat's color, freshness, and overall quality [42, 43]”.

Grape Residue Analysis

Question why the fatty acids profile in grape residue was determined but not the antioxidant capacity or total polyphenol content. Suggest linking these unexplored areas to a comprehensive analysis of the raw feed material.

Response: Thanks for your good question. Because of the reported associations between dietary fat consumption and health outcomes including CVD and cancer, consumers are increasingly taking into consideration the fat and FA composition of meat products when making purchasing decisions[2,3]. Therefore, current efforts to enhance the health value of beef lipids have focused on feeding strategies that reduce the meat content of SFA (e.g., 16:0) and specific trans-FA (TFA; e.g., 18:1 t-10), and increase the content of specific CLA (e.g., 18:2 c9t11) and TFA (e.g., 18:1 t-11) isomers and n-3 and n-6 PUFA (e.g., 18:3n-3 and 18:2 n-6) [2,3]. Therefore, we chose to determine the fatty acid content of grape pomace, but not the antioxidant capacity or the total polyphenol content. In the subsequent experiments, we will determine the total polyphenol content in grape pomace and extracted procyanidins with potent antioxidant effects, and further investigated its antioxidant capacity.

Comments on the Quality of English Language

Minor typos and misspelled words.

Response: Thank you very much for the reminder. We have made revisions accordingly.

References

  1. Chikwanha, O.C.; Muchenje, V.; Nolte, J.E.; Dugan, M.; Mapiye, C. Grape pomace (Vitis vinifera L. cv. Pinotage) supplementation in lamb diets: Effects on growth performance, carcass and meat quality. Meat Sci. 2019, 147, 6-12, doi: 10.1016/j.meatsci.2018.08.017.
  2. Scollan, N.D.; Price, E.M.; Morgan, S.A.; Huws, S.A.; Shingfield, K.J. Can we improve the nutritional quality of meat? Proc Nutr Soc 2017, 76, 603-618, doi: 10.1017/S0029665117001112.
  3. Wang, D.D.; Hu, F.B. Dietary Fat and Risk of Cardiovascular Disease: Recent Controversies and Advances. Annu. Rev. Nutr. 2017, 37, 423-446, doi: 10.1146/annurev-nutr-071816-064614.

Sincerely,

Bo Yao

Reviewer 2 Report

Comments and Suggestions for Authors

The manuscript titled “Improving Shelf-life and Content of Unsaturated Fatty Acids in Meat of Lambs by Supplemented with Grape Dregs in Diet” was aimed to improve the quality of lamb meat, extend the shelf life of meat and the composition as well as proportion of fatty acids or CLA by adding grape residue in fattening lamb’s diets. The authors also investigated the potential effects of grape dregs on the expression of CLA deposits-related genes such as SCD, PPARα and PPARγ in the lamb tissues, which opens a new direction of considering, processing and application of grape pomace in food and feed technologies. The work is interesting and deserves consideration. However, the at presented state the manuscript needs revision. The main comments and recommendations are listed below.

1.       I suggest to revise the title to make it shorter and stylistically correct

2.       The authors used GP abbreviation in the Abstract without definition. The rest abbreviations are not needed in the Abstract. Moreover, the authors can refuse the use of abbreviations in the Abstract. This will be the best decision.

3.       Key words just repeat the title. It is recommended to use other key words to expand potential of the article to be found by readers

4.       L. 36-37. This sentence confuses. The authors write that mutton has low fat and cholesterol content. This is not correct. Approximate fat content in mutton is 20%. Reference [1] did not confirm low fat and cholesterol content as well.

5.       L. 45-49. This sentence can be presented simpler. Too long sentence

6.       L. 50. It is better to use linking sentence to make a smooth cross between paragraphs 1 and 2.

7.       Paragraphs 2 and 3 can be reached by more references with related works. For example, a briefly search revealed the work on reduction of oxidative activity in raw beef meat using grape pomance, or usage of BAS of grape in technology of meat products:

Sadovoy, V. V.; Selimov, M.A.;  Shchedrina, T.V.; Nagdalian, A.A. Usage of biological active supplements in technology of prophilactic meat products. Research Journal of Pharmaceutical, Biological & Chemical Sciences.2016, 7, 1861–65.

Bennato, F.; Martino, C.; Ianni, A.; Giannone, C.; Martino, G. Dietary Grape Pomace Supplementation in Lambs Affects the Meat Fatty Acid Composition, Volatile Profiles and Oxidative Stability. Foods 202312, 1257. https://doi.org/10.3390/foods12061257

8.       L. 92. It is better to make a new paragraph

9.       Table 1. The authors used abbreviations without definition

10.   Materials and methods. The authors should give details for all equipment, chemicals and software used in the experiment. Usually presented as …model (Manufacturer, City, Country)

11.   Why the authors used longissimus dorsi in subsection 2.3.2, but latissimus dorsi in subsection 2.3.4.?

12.   Subsection 2.4. What p-value was set by the authors as valuable?

13.   Conclusion should be reached by the most important results (data) obtained.

14.   References have duplicated order numbers

15.   The manuscript should be checked by native English speaker for typos and grammatical errors corrections

Comments on the Quality of English Language

The manuscript should be checked by native English speaker for typos and grammatical errors corrections

Author Response

Dear Editor,

We appreciate you and the reviewers for your precious time in reviewing our paper and providing valuable comments. It was your valuable and insightful comments that led to possible improvements in the current version. The authors have carefully considered the comments and tried our best to address every one of them. We hope the manuscript after careful revisions meet your high standards. The authors welcome further constructive comments if any. Below we provide the point-by-point responses. All modifications in the manuscript have been highlighted in yellow.

Response to Reviewer 2

[General Comment] The manuscript titled “Improving Shelf-life and Content of Unsaturated Fatty Acids in Meat of Lambs by Supplemented with Grape Dregs in Diet” was aimed to improve the quality of lamb meat, extend the shelf life of meat and the composition as well as proportion of fatty acids or CLA by adding grape residue in fattening lamb’s diets. The authors also investigated the potential effects of grape dregs on the expression of CLA deposits-related genes such as SCD, PPARα and PPARγ in the lamb tissues, which opens a new direction of considering, processing and application of grape pomace in food and feed technologies. The work is interesting and deserves consideration. However, the at presented state the manuscript needs revision. The main comments and recommendations are listed below.

Response: Thank you very much for recognizing our original intent in writing this manuscript. We have read your comments carefully and tried our best to address them one by one. It can be seen in line 34 to line 476.

  1. I suggest to revise the title to make it shorter and stylistically correct

Response: Thank you very much for the reminder. We have made revisions accordingly. The title has revise as “Improving Shelf-life and Content of Unsaturated Fatty Acids in Lambs Meat Diet Supplemented with Grape Dregs”.

  1. The authors used GP abbreviation in the Abstract without definition. The rest abbreviations are not needed in the Abstract. Moreover, the authors can refuse the use of abbreviations in the Abstract. This will be the best decision.

Response: Thank you for your nice reminder. We have made revisions accordingly. It can be seen in line 23.

  1. Key words just repeat the title. It is recommended to use other key words to expand potential of the article to be found by readers.

Response: Thank you for your nice reminder. We've changed the keywords of the article as “grape pomace, meat production, shelf life and CLA deposits-related genes”.

  1. L. 36-37. This sentence confuses. The authors write that mutton has low fat and cholesterol content. This is not correct. Approximate fat content in mutton is 20%. Reference [1] did not confirm low fat and cholesterol content as well.

Response: Thank you for your nice reminder. We have changed " Mutton as a meat is increasingly recognized and preferred by people because of its high protein, low fat and cholesterol content" to " The nutritional richness of mutton, including its high content of protein, iron, zinc, and vitamin B, is gaining recognition and preference among people " in this paper and re-cited the literature.

  1. L. 45-49. This sentence can be presented simpler. Too long sentence.

Response: Thank you for your nice reminder. We have made revisions accordingly. It can be seen in line 47 to line 49.

  1. L. 50. It is better to use linking sentence to make a smooth cross between paragraphs 1 and 2.

Response: Thank you for your nice reminder. We have deleted the sentence " The composition of grape pomace (GP) includes grape seeds, grape skins, and grape stalks. The skins and seeds of grapes are abundant in diverse polyphenols such as flavonoids, tannins, and anthocyanins." from the article to make paragraphs 2 and 3 more logically coherent.

  1. Paragraphs 2 and 3 can be reached by more references with related works. For example, a briefly search revealed the work on reduction of oxidative activity in raw beef meat using grape pomance, or usage of BAS of grape in technology of meat products:

Sadovoy, V. V.; Selimov, M.A.;  Shchedrina, T.V.; Nagdalian, A.A. Usage of biological active supplements in technology of prophilactic meat products. Research Journal of Pharmaceutical, Biological & Chemical Sciences.2016, 7, 1861–65.

Bennato, F.; Martino, C.; Ianni, A.; Giannone, C.; Martino, G. Dietary Grape Pomace Supplementation in Lambs Affects the Meat Fatty Acid Composition, Volatile Profiles and Oxidative Stability. Foods 2023, 12, 1257. https://doi.org/10.3390/foods12061257.

Response: Thank you for your nice reminder. We have made revisions accordingly. It can be seen in line 51 and 64.

  1. L. 92. It is better to make a new paragraph.

Response: We have made revisions accordingly. It can be seen in line 82.

9.Table 1. The authors used abbreviations without definition.

Response: Thank you for your nice reminder. We have made revisions accordingly. It can be seen in line 106 to line111.

  1. Materials and methods. The authors should give details for all equipment, chemicals and software used in the experiment. Usually presented as …model (Manufacturer, City, Country).

Response: Thanks for your kind reminders. We have added in materials and methods of the text. It can be seen in line 125, 129, 151, 152,155, 160 and 180.

  1. Why the authors used longissimus dorsiin subsection 2.3.2, but latissimus dorsiin subsection 2.3.4.?

Response: Thank you for the nice reminder. It should be the longissimus dorsi in the entire text. We have changed " latissimus dorsi " to " longissimus dorsi " in 2.3.4.

  1. Subsection 2.4. What p-value was set by the authors as valuable?

Response: Thank you for your nice reminder. We have made revisions accordingly. We consider P<0.05 in the text to indicate a significant difference.

  1. Conclusion should be reached by the most important results (data) obtained.

Response: Thank you for your nice reminder. We have made revisions as to “The inclusion of 16% and 24% grape dregs in the diet notably extended the shelf life of lamb meat. Specifically, the presence of 16% grape dregs significantly boosted Conjugated Linoleic Acid (CLA) levels and the concentrations of various unsaturated fatty acids (C18:1n-9c, C18:2n-6c, C20:3n-6, C20:4n-6, C22:0, C22:1n-9, C22:2), as well as saturated fatty acids (C23:0 and C24:0) in the longissimus dorsi muscle of the lambs. This increase ultimately led to a significant rise in the ∑CLA/Total Fatty Acid (TFA) ratio.”.

  1. References have duplicated order numbers.

Response: We have made revisions accordingly.

15.The manuscript should be checked by native English speaker for typos and grammatical errors corrections.

Response: We read through the entire manuscript to eliminate misspellings and grammatical errors.

Comments on the Quality of English Language

The manuscript should be checked by native English speaker for typos and grammatical errors corrections.

Response: Thank you for your nice reminder. We have made revisions accordingly.

Sincerely,

Bo Yao

Reviewer 3 Report

Comments and Suggestions for Authors

Some adjustments needed.

-       Respect the journal's instructions for references. 

-       In table 1 indicate the for levels instead of A … D

-       Correct the many typos.

-       In table 4 replace GR with “Grow Rate” and center “GP levels” between levels.

-       Verify tu use the same font along the test.

Author Response

Dear Editor,

We appreciate you and the reviewers for your precious time in reviewing our paper and providing valuable comments. It was your valuable and insightful comments that led to possible improvements in the current version. The authors have carefully considered the comments and tried our best to address every one of them. We hope the manuscript after careful revisions meet your high standards. The authors welcome further constructive comments if any. Below we provide the point-by-point responses. All modifications in the manuscript have been highlighted in yellow.

Response to Reviewer 3

-Respect the journal's instructions for references. 

Response: Thank you for your comments. We have made revisions accordingly.

-In table 1 indicate the for levels instead of A … D

Response: Thank you for your comments. We have made revisions accordingly.

- Correct the many typos.

Response: We read through the entire manuscript to eliminate misspellings.

- In table 4 replace GR with “Grow Rate” and center “GP levels” between levels.

Response: Thank you for your comments. The GR value refers to the thickness of the tissue between the 12th and 13th ribs, 11 cm from the midline of the dorsal spine, and represents an indication of the fat content of the carcass. The articles usually write the GR values directly [1].

- Verify tu use the same font along the test.

Response: Thank you for your comments. We read through the entire manuscript to ensure that the same font is used throughout.

References

1.Chen, X.; Mi, H.; Cui, K.; Zhou, R.; Tian, S.; Zhang, L. Effects of Diets Containing Finger Millet Straw and Corn Straw on Growth Performance, Plasma Metabolites, Immune Capacity, and Carcass Traits in Fattening Lambs. Animals (Basel) 2020, 10, doi: 10.3390/ani10081285.

Sincerely,

Bo Yao

Round 2

Reviewer 1 Report

Comments and Suggestions for Authors

The authors responded very well and concise to the previous observations. 

They have corrected the paper in a very nice manner and enhanced it's academic quality. 

I have no further observations. 

Reviewer 2 Report

Comments and Suggestions for Authors

The authors considered all.commenrs and recommendations and decided them well. The revised manuscript can be recommended for publication